# OOVDet: Low-Density Prior Learning for Zero-Shot Out-of-Vocabulary Object Detection

Binyi Su [1 2]  Chenghao Huang [1]  Haiyong Chen [1]

## Abstract

Zero-shot out-of-vocabulary detection (ZS-OOVD) aims to accurately recognize objects of in-vocabulary (IV) categories provided at zero-shot inference, while simultaneously rejecting undefined ones (out-of-vocabulary, OOV) that lack corresponding category prompts. However, previous methods tend to overfit IV classes, resulting in undefined OOV objects being confidently misclassified as semantically similar IV categories. To mitigate this issue, this paper proposes a zero-shot OOV detector (OOVDet), a novel framework that effectively detects predefined classes while reliably rejecting undefined ones in zero-shot scenes. Specifically, due to the model's lack of prior knowledge about the distribution of OOV data, we synthesize region-level OOV prompts by sampling from the low-likelihood regions of the class-conditional Gaussian distributions in the hidden space, motivated by the assumption that unknown semantics are more likely to emerge in low-density areas of the latent space. For OOV images, we further propose a Dirichlet-based gradient attribution mechanism to mine pseudo-OOV image samples, where the attribution gradients are interpreted as Dirichlet evidence to estimate prediction uncertainty, and samples with high uncertainty are selected as pseudo-OOV images. Building on these synthesized OOV prompts and pseudo-OOV images, we construct the OOV decision boundary through a low-density prior constraint, which regularizes the optimization of OOV classes using Gaussian kernel density estimation in accordance with the above assumption. Experimental results show that our method significantly improves the OOV detection performance in zero-shot scenes. The code is available at https://github.com/binyisu/OOVDet.

[1]School of Artificial Intelligence and Data Science, Hebei University of Technology, Tianjin, China. [2]China Xiongan Group Digital City Technology Company Ltd., Xiongan, China. Correspondence to: Haiyong Chen <haiyong.chen@hebut.edu.cn>.

*Proceedings of the 43rd International Conference on Machine Learning*, Seoul, South Korea. PMLR 306, 2026. Copyright 2026 by the author(s).

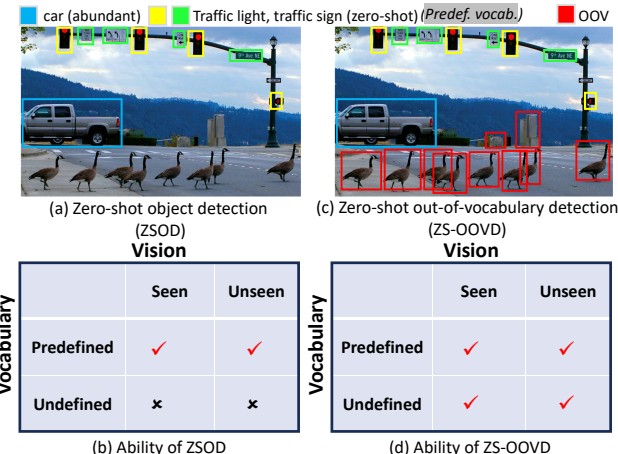

Figure 1. The zero-shot object detection (ZSOD) task vs the zero-shot out-of-vocabulary detection (ZS-OOVD) task.

## 1. Introduction

Zero-shot object detection (ZSOD) (Bansal et al., 2018; Cheng et al., 2024) has garnered significant attention for its ability to detect unseen categories at inference. The integration of vision-language models (Radford et al., 2021; Zhang et al., 2025a) and prompt learning (Zhou et al., 2022b; 2026; Li et al., 2026) has advanced ZSOD (Qutub et al., 2024; Guo et al., 2025) and open-vocabulary object detection (OVOD) (Minderer et al., 2022; 2023). However, in real-world scenarios, numerous objects fall outside the predefined vocabulary, as illustrated in Fig. 1 (c). For instance, applying a vocabulary such as ['car', 'traffic light', 'traffic sign'] to the image in Fig. 1 (c) may lead to the misdetection or omission of objects not covered by the predefined categories. This highlights a critical challenge: how can a zero-shot model accurately identify objects within the predefined vocabulary while reliably rejecting objects from undefined categories?

In practice, this problem is more challenging than conventional ZSOD. As illustrated in Fig. 1(d), we define a new task, termed zero-shot out-of-vocabulary object detection (ZS-OOVD), which requires rejecting semantic concepts beyond a predefined vocabulary. In contrast to ZSOD's visual-centric treatment of unknowns, ZS-OOVD adopts a language-centric perspective with an explicit focus on vocabulary boundaries. Despite its practical importance, this problem has not been explicitly formulated or systematically studied, as existing ZSOD approaches focus on detecting

unseen classes within a fixed vocabulary while largely ignoring continuously emerging out-of-vocabulary objects in open-world scenarios. This gap highlights the necessity of investigating ZS-OOVD, particularly for safety-critical applications. Motivated by this gap, we take a first step toward defining the ZS-OOVD task, establishing a benchmark, and proposing an effective framework, termed OOVDet, which achieves reliable OOV detection while maintaining high accuracy on predefined categories.

Specifically, existing models lack prior knowledge of OOV data distributions, including OOV prompts and images. Due to the absence of these categories in training, models lack both semantic and visual priors, resulting in weak decision boundaries and unreliable predictions for OOV objects. To mitigate these issues, we first introduce a region-level Out-of-vocabulary Prompt Synthesis (OPS) mechanism that samples from the low-likelihood regions of IV class-conditional Gaussian distributions in the hidden space, motivated by the assumption that OOV semantics are more likely to emerge in low-density areas of the latent manifold. For the IV data, we first provide predefined question–answer (Q&A) pairs and templates that incorporate category names, for example: **Q:** `[CLS] What object can be seen in the region?` **A:** `[CLS] The object is <cls>`. We then guide ChatGPT through multiple rounds of prompt standardization to construct an IV class-specific prompt library, sampled during data preparation. This prompts are encoded by a text encoder to obtain embeddings, followed by the addition of masked Gaussian noise to form an IV class-conditional Gaussian distribution. This distribution is then used to synthesize OOV prompt embeddings in the latent space. Additionally, owing to the absence of training images for OOV classes, we select high-uncertainty samples from region proposals and treat them as pseudo-OOVs for optimization. Unlike prior energy-based mining methods (Joseph et al., 2021; Su et al., 2024), we propose a Dirichlet-based Gradient Attribution (DGA) approach that models attribution gradients as Dirichlet evidence to estimate predictive uncertainty and mine high-uncertainty samples as pseudo-OOV instances.

In particular, given the observation that in-vocabulary objects cluster in high-density regions while out-of-vocabulary objects reside in low-density areas (Han et al., 2022; Ren et al., 2018), proper separation of these regions is essential for reliable OOV decision boundaries. Prior methods (Han et al., 2022) expand low-density regions via contrastive learning but do not explicitly enforce density separation. In contrast, we adopt a density estimation strategy, using Gaussian kernel density estimation to push pseudo-OOV samples into low-density regions and away from dense IV clusters. Specifically, we propose a Low-density Prior Constraint (LPC) to regularize OOV optimization and promote discriminative IV-OOV decision boundaries.

In summary, the contribution of this paper is three-fold:

- To the best of our knowledge, this is the first work to confront the challenging ZS-OOVD task, which achieves effective OOV rejection without degrading performance on the predefined closed-set vocabulary.
- We propose a novel OOV detector, termed OOVDet, which integrates three well-designed modules-OPS, DGA, and LPC-to enable robust and reliable IV-OOV discrimination.
- We introduce a new ZS-OOVD benchmark. Compared to previous methods, our approach consistently achieves substantially higher recall for OOV classes, providing 4.77 - 8.70% improvements across different settings of OOV datasets.

## 2. Related Works

### 2.1. Out-of-Vocabulary Detection

Out-of-vocabulary (OOV) detection was originally studied in speech recognition (Young, 1994) and has recently attracted attention in computer vision with the emergence of vision-language models and prompt learning. However, OOV detection in zero-shot settings remains largely underexplored. Unlike conventional zero-shot or open-vocabulary detection (Zhong et al., 2022), zero-shot OOV detection lacks both visual exemplars and reliable linguistic priors for OOV categories, making accurate recognition and localization particularly challenging. Recent studies have attempted to mitigate unknown modeling by introducing pseudo-sample generation or proxy supervision (Wu et al., 2025; Su et al., 2023). Inspired by this line of work, we propose an OOV prompt synthesis and pseudo-OOV image mining framework that compensates for the absence of real OOV supervision, alleviates prior deficiency.

### 2.2. Zero-Shot and Open-Vocabulary Object Detection

Zero-shot and open-vocabulary object detection leverage large-scale image-text pretraining to recognize unseen categories via visual-textual alignment (Bansal et al., 2018; Zhong et al., 2022; Minderer et al., 2022). While effective for unseen but predefined classes, these methods are restricted to IV prediction and often misclassify undefined OOV objects into semantically similar IV categories with high confidence. From a representation perspective, supervised learning induces compact, high-density IV clusters, whereas OOV samples tend to occupy low-density regions between clusters. Density-aware methods such as OpenDet (Han et al., 2022) and AKCR (Sarkar et al., 2024) exploit this property via feature compactness or semantic alignment, but depend on unknown-class supervision or are unsuitable for zero-shot OOV detection, leading to limited OOV rejection. To overcome this limitation, we propose a low-density prior learning method that constrains pseudo-OOV priors in low-density regions using Gaussian kernel density estimation, thereby constructing discriminative IV-OOV boundaries without real OOV supervision, and improving OOV detection performance in a zero-shot setting.

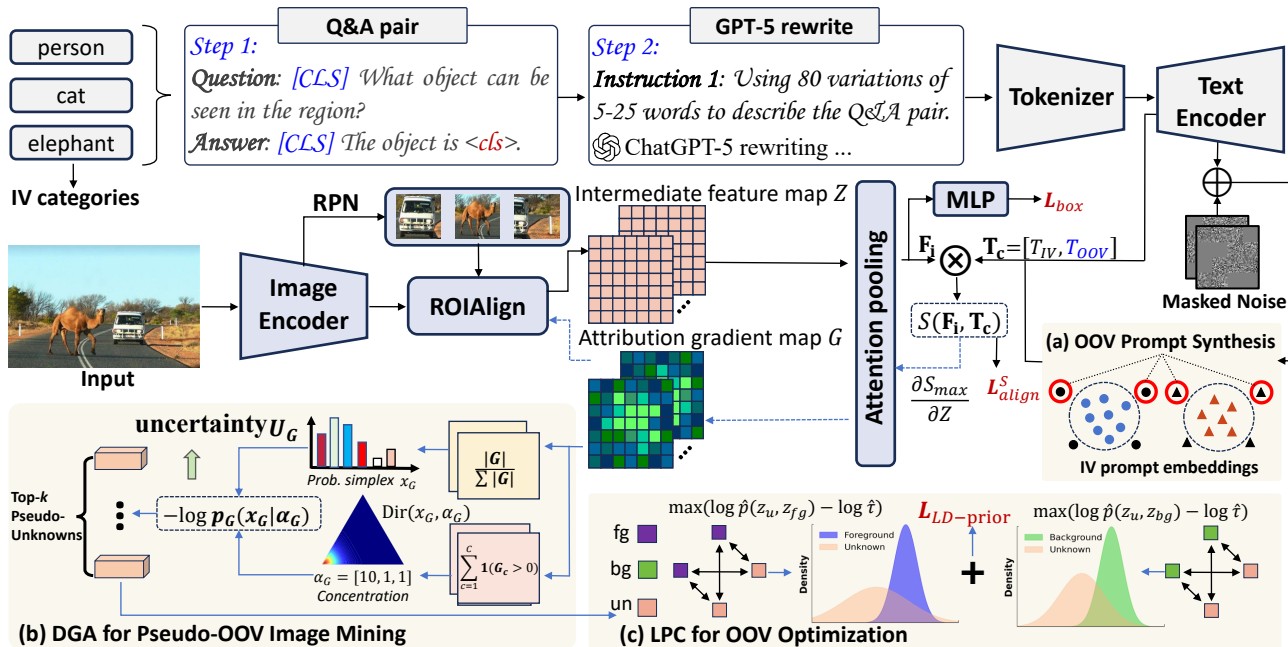

Figure 2. The framework of our OOVDet for the challenging zero-shot out-of-vocabulary object detection task. OOVDet is a simple two-stage detector with (a) a novel Out-of-vocabulary Prompt Synthesis (OPS), (b) a novel Dirichlet-based Gradient Attribution (DGA) for pseudo-unknowns mining and (c) a novel Low-density Prior Constraint (LPC) for OOV optimization.

## 3. Method

We define the ZS-OOVD problem setup with reference to ZSOD task (Bansal et al., 2018) and OOV detection task (Young, 1994). As illustrated in Fig. 3, given an object detection dataset $D = (x, y)$, where $x \in \mathbf{X}$ denotes an input image and $y = \{(c_i, \hat{b}_i)\}_{i=1}^{I}$ represents objects with their class label $c_i$ and bounding box $\hat{b}_i$, the dataset is divided into a training set $D_{tr}$ and a testing set $D_{te}$. The training set $D_{tr}$ contains $S$ seen classes denoted as $C_{Seen}$, with predefined vocabularies. In practice, the testing set $D_{te}$ includes the category set $C_T = C_{Seen} \cup C_{Unseen} \cup C_{OOV}$, where $C_{Unseen}$ denotes $Z$ predefined unseen classes, and $C_{OOV}$ represents undefined classes that are entirely unknown to the model. By definition, $(C_{Seen} \cup C_{Unseen}) \cap C_{OOV} = \varnothing$. Given the unbounded number of unknown categories in open-world settings, we consolidate them into a single "OOV" class. Accordingly, the objective is to train a detector on $D_{tr}$ that jointly recognizes $S$ seen classes, $Z$ unseen classes, and one OOV class under an open-set assumption. Here we let $K = S + Z$. Accordingly, the final scores are predicted as: scores = [$\text{seen}_1, \ldots, \text{seen}_S, \text{unseen}_1, \ldots, \text{unseen}_Z, \text{OOV}$].

### 3.1. Baseline Setup

We setup the baseline with RegionClip (Zhong et al., 2022), which consists of a image encoder, a text encoder, a separately trained region proposal network (RPN), and an R-CNN. Conventional prompt learning aligns each proposal feature $\mathbf{F}_i$ with its corresponding class prompt $\mathbf{T}_c$ by com-

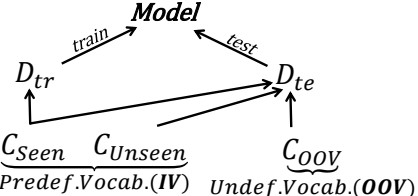

Figure 3. Problem Setup of ZS-OOVD.

puting their cosine similarity $S(\mathbf{F}_i, \mathbf{T}_c)$, which serves as the semantic alignment objective for category optimization:

$$\boldsymbol{L}_{align}^{S} = -\frac{1}{N} \sum_{i=1}^{N} \sum_{j=1}^{K+2} y_{ij} \log \frac{\exp(S(\mathbf{F}_i, \mathbf{T}_j)/\tau)}{\sum_{c=1}^{K} \exp((S(\mathbf{F}_i, \mathbf{T}_c))/\tau)}, \quad (1)$$

where $S(\cdot, \cdot)$ represents the cosine similarity and $\tau$ denotes the temperature parameter, $y_{ij}$ is an indicator (0 or 1) of sample $i$ belonging to category $j$ in the ground truth label. Compared with the RegionClip, we propose an Out-of-vocabulary Prompt Synthesis (OPS) for text embedding representation of undefined vocabulary, a Dirichlet-based Gradient Attribution (DGA) module for pseudo-OOV image mining, a Low-density Prior Constraint (LPC) loss for OOV optimization, as shown in Fig. 2.

For IV categories, ChatGPT-5 is used to rewrite predefined Q&A pairs, generating multiple prompt variants to form a class-specific prompt library, where the <cls> token is replaced with the corresponding category label. These prompts are fed into the text encoder, resulting in a bag-of-words tensor $\mathbf{B}$. After averaging by category, we obtain

the final IV prompt representation $\mathbf{T}_{IV}$. For OOV categories, random masked Gaussian noise is added to each IV prompt embedding in $\mathbf{B}$, forming a class-conditional Gaussian distribution. Then, this distribution is used to synthesize the virtual OOV prompt representation $\mathbf{T}_{OOV}$ within the low-likelihood region of the IV class-conditional Gaussian distribution in the latent space. The prompt embedding of all categories is given as $\mathbf{T}_c = \{\mathbf{T}_{IV}, \mathbf{T}_{OOV}\}$.

## 3.2. Out-Of-Vocabulary Prompt Synthesis

Prior methods (Wu et al., 2025; Su et al., 2024) focus on visual feature mining and ignore textual OOV prompts, limiting explicit modeling of IV-OOV distributional gaps. To address this, we separate IV and OOV prompts in latent space and generate region-level pseudo-OOV prompts via Gaussian outlier sampling from low-likelihood regions of class-conditional IV Gaussians, enabling contrastive text–image alignment across IV and OOV. Assuming sufficient data, IV prompt embeddings from the text encoder are modeled as class-conditional multivariate Gaussian distributions:

$$p_\theta(\vec{T} \mid \vec{y} = i) = \mathcal{N}(\vec{\mu}_i, \hat{\sigma}), \tag{2}$$

where $\theta$ denotes the parameters of the text encoder $f_\theta$, $\vec{y}$ is the ground-truth label, $\vec{\mu}_i$ is the empirical Gaussian mean of the $i$-th IV class prompts embedding, and $i \in \{1, \ldots, K\}$ with $K$ being the number of IV classes (seen + predefined unseen). The Gaussian distribution is defined as:

$$\mathcal{N}(\vec{\mu}_i, \hat{\sigma}) = \frac{1}{\sqrt{2\pi\hat{\sigma}^2}} \exp\left(-\frac{(\vec{T} - \vec{\mu}_i)^2}{2\hat{\sigma}^2}\right),$$

where $\hat{\sigma}$ is the tied covariance matrix across all IV classes.

The empirical mean of the $i$-th IV class prompts embedding is computed as:

$$\vec{\mu}_i = \frac{1}{|\mathcal{Q}_T|} \sum_{j=1}^{|\mathcal{Q}_T|} \vec{T}_{i,j}, \tag{3}$$

where $|\mathcal{Q}_T|$ is the size of the prompts queue $\mathcal{Q}_T \in \mathbb{R}^{K \times |\mathcal{Q}_T|}$. The tied covariance matrix is estimated as:

$$\hat{\sigma} = \frac{1}{K|\mathcal{Q}_T|} \sum_{i=1}^{K} \sum_{j=1}^{|\mathcal{Q}_T|} (\vec{T}_{i,j} + \alpha\mathbf{m} \odot \vec{\varepsilon} - \vec{\mu}_i) \cdot (\vec{T}_{i,j} + \alpha\mathbf{m} \odot \vec{\varepsilon} - \vec{\mu}_i)^T + \beta\mathbf{E}, \tag{4}$$

where $\mathbf{m} \sim \text{Bernoulli}(\cdot)$ is a random mask vector, $\vec{\varepsilon}$ is a learnable noise matrix following a Gaussian distribution, $\mathbf{E}$ is the identity matrix, and $\alpha, \beta$ are hyperparameters. Here, the masked noise stabilizes $\hat{\sigma}$ by preventing degeneration and ensuring a well-conditioned inverse.

Using these estimated Gaussian parameters, the prompt synthesis module samples virtual OOV prompts from the low-likelihood regions of each IV class distribution:

$$\vec{v}_i \in \Psi(\vec{T}, \vec{\mu}_i, \hat{\sigma}), \tag{5}$$

where $\Psi$ denotes the class-conditional Gaussian distribution probability density. The joint probability across all IV classes is:

$$\Psi(\vec{T}, \vec{\mu}_1, \ldots, \vec{\mu}_K, \hat{\sigma}) = \prod_{i=1}^{K} \Psi(\vec{T}, \vec{\mu}_i, \hat{\sigma}). \tag{6}$$

For each $\Psi(\vec{T}, \vec{\mu}_i, \hat{\sigma})$, the low-likelihood region is defined as:

$$\Psi(\vec{T}, \vec{\mu}_i, \hat{\sigma}) = \left\{\vec{v}_i \sim \mathcal{N}(\vec{\mu}_i, \hat{\sigma}) \,\Big|\, \frac{1}{(2\pi)^{d/2}|\hat{\sigma}|^{1/2}} \cdot \exp\left[-\frac{1}{2}(\vec{v}_i - \vec{\mu}_i)^T \hat{\sigma}^{-1}(\vec{v}_i - \vec{\mu}_i)\right] < \varepsilon\right\}, \tag{7}$$

where $\vec{v}_i$ is a virtual OOV prompt representation sampled from the $i$-th IV class, $d$ is the embedding dimension, and $\varepsilon$ is a threshold controlling the low-likelihood region. Since the exponent term in Eq. (7), $D_i(\vec{v}_i) = -\frac{1}{2}(\vec{v}_i - \vec{\mu}_i)^\top \hat{\sigma}^{-1}(\vec{v}_i - \vec{\mu}_i)$ is proportional to the Mahalanobis distance, samples with larger distances lie in lower-likelihood regions of the IV distribution. In practice, we rank all perturbed candidates by their Mahalanobis distance and take the lowest-likelihood sample as the synthesized OOV prompt embedding $\mathbf{T}_{OOV}$.

## 3.3. Dirichlet-based Gradient Attribution for Pseudo-OOV image Mining

The lack of real OOV images during training hinders reliable OOV boundary construction, and prior methods (Han et al., 2022; Su et al., 2023) mainly mine high-uncertainty pseudo-unknown samples using logit-based metrics which may not fully capture deeper uncertainty. Motivated by attribution-based interpretability methods (Simonyan et al., 2013; Selvaraju et al., 2017), we model intermediate-layer gradients using a generalized Dirichlet distribution. As illustrated in Fig. 2(b), the attribution gradient $G$ is defined as the partial derivative of the classification similarity $S$ with respect to the layer parameters $Z$. Analogous to Grad-CAM (Selvaraju et al., 2017), we select the three lowest-density foreground proposals under the Dirichlet model as pseudo-OOV candidates. In Dirichlet distribution, the probability density of the gradient map can be expressed as:

$$\log p_G(x_G^n \mid \alpha_G^n) = \underbrace{\sum_{c=1}^{C}(\alpha_G^{n,c} - 1)\log x_G^{n;c}}_{\text{Data term}} - \underbrace{\left(\sum_{c=1}^{C}\log\Gamma(\alpha_G^{n;c}) - \log\Gamma\left(\sum_{c=1}^{C}\alpha_G^{n,c}\right)\right)}_{\text{Normalization term}}, \tag{8}$$

where $\Gamma$ is the Gamma function used to calculate the normalization coefficients of the Dirichlet distribution, $n = 1, 2, \ldots, N$ indexes each proposal. $x_G$ and $\alpha_G$ denote the probability simplex and the concentration in Dirichlet function, respectively.

Since each proposal yields $C$ gradient maps and the attribution gradients are assumed to follow a generalized Dirichlet distribution ($x_G \sim \mathrm{Dir}(x_G, \alpha_G)$), jointly estimating $\alpha_G$ is computationally expensive. We therefore approximate $\alpha_G$ using the frequency of positive gradients across channels, preserving essential distributional cues while reducing computation. The probability simplex $x_G$ and the concentration $\alpha_G$ can be jointly expressed by:

$$
\begin{aligned}
x_G^{n,c} &= \left| \frac{\partial S_{max}^n}{\partial Z_{n,c}} \right| \bigg/ \sum_{c=1}^{C} \left| \frac{\partial S_{max}^n}{\partial Z_{n,c}} \right|, \\
\alpha_G^{n,c} &= \sum_{w=1}^{W} \sum_{h=1}^{H} \mathbf{1} \left( \frac{\partial S_{max}^n}{\partial Z_{n,c,w,h}} > 0 \right),
\end{aligned}
\tag{9}
$$

where $S_{max}$ denotes the maximum classification similarity. As shown in Fig. 2 (b), we quantify the uncertainty $\mathcal{U}_G$ of each proposal using the negative log-probability density of the gradient distribution, and leverage the resulting high-uncertainty pseudo-OOV instances to enhance the OOV-class modeling:

$$
U_G = -\log(p_G(x_G \mid \alpha_G)). \tag{10}
$$

Furthermore, since background proposals often overwhelm the mini-batch, we adopt balanced sampling with equal numbers of foreground and background proposals, which improves the model's ability to recall OOV objects appearing in the background class.

### 3.4. Low-Density Prior Constraint for OOV Optimization

In supervised learning, IV-class samples are tightly aligned and form compact, high-density clusters in the embedding space. OOV samples, lacking such alignment, naturally reside in the low-density regions between these clusters. This distributional gap motivates our Low-Density Prior Constraint, which explicitly uses these low-density regions as priors to guide OOV optimization. To learn pseudo-OOV characteristics, we introduce a placeholder outside the existing vocabulary to represent OOV classes and leverage the low-density prior to model IV-OOV relations. Given pseudo-OOV samples $z_u \in \mathbb{R}^{A \times d}$ and an IV feature bank $z_{\mathrm{IV}} = \{z_1, \ldots, z_N\} \in \mathbb{R}^{N \times d}$, we estimate the density between each OOV sample and $z_{\mathrm{IV}}$ using a Gaussian kernel:

$$
\hat{p}(z_u^{(j)}) = \frac{1}{N} \sum_{i=1}^{N} \frac{1}{(2\pi h^2)^{d/2}} \exp \left( -\frac{\|z_u^{(j)} - z_i\|^2}{2h^2} \right), \tag{11}
$$

where $A$ is the number of pseudo-OOV samples, $h$ is the kernel bandwidth, and $d$ is the feature dimension. For each pseudo-OOV sample, we enforce a low-density prior on pseudo-OOVs via their log-density and apply it separately to foreground and background proposals.

**Foreground pseudo-OOVs.** The low-density constraint for foreground pseudo-OOV proposal features is defined as:

$$
\mathcal{L}_{\mathrm{LD}}^{\mathrm{fg}} = \frac{1}{B_{\mathrm{fg}}} \sum_{j=1}^{B_{\mathrm{fg}}} \underbrace{s_j^{\mathrm{fg}} \big(1 - s_j^{\mathrm{fg}}\big)^{\alpha}}_{t_j^{\mathrm{fg}}} \cdot \max \big( \log \hat{p}(z_{u,\mathrm{fg}}^{(j)}) - \log \hat{\tau}, \, 0 \big),
\tag{12}
$$

where $z_{u,\mathrm{fg}}^{(j)}$ denotes the $j$-th foreground pseudo-OOV proposal feature, $s_j^{\mathrm{fg}}$ is the ground-truth probability for the $j$-th foreground sample, and $B_{\mathrm{fg}}$ is the number of foreground pseudo-OOV samples. $t_j^{\mathrm{fg}}$ is the corresponding weighting factor. $\hat{\tau}$ donates the density threshold.

**Background pseudo-OOVs.** Refer to the definitions of all parameters in the low-density constraint for foreground pseudo-OOV proposal features (Section "Foreground pseudo-OOVs"). The low-density constraint for background pseudo-OOV samples is given by:

$$
\mathcal{L}_{\mathrm{LD}}^{\mathrm{bg}} = \frac{1}{B_{\mathrm{bg}}} \sum_{j=1}^{B_{\mathrm{bg}}} \underbrace{s_j^{\mathrm{bg}} \big(1 - s_j^{\mathrm{bg}}\big)^{\alpha}}_{t_j^{\mathrm{bg}}} \cdot \max \big( \log \hat{p}(z_{u,\mathrm{bg}}^{(j)}) - \log \hat{\tau}, \, 0 \big).
\tag{13}
$$

**Total low-density prior constraint.** The overall loss combining foreground and background components is:

$$
\mathcal{L}_{\mathrm{LD\text{-}prior}} = \mathcal{L}_{\mathrm{LD}}^{\mathrm{fg}} + \mathcal{L}_{\mathrm{LD}}^{\mathrm{bg}}, \tag{14}
$$

where $\mathcal{L}_{\mathrm{LD}}^{\mathrm{fg}}$ and $\mathcal{L}_{\mathrm{LD}}^{\mathrm{bg}}$ denote the low-density losses for foreground and background pseudo-OOV samples, respectively. This formulation penalizes pseudo-OOVs only in high-density IV regions, preventing their collapse into IV clusters and sharpening the OOV boundary. As Fig. 6 (b) shows, IV samples form compact high-density clusters, while OOVs lie in the low-density gaps between them, modeling these gaps as priors guides pseudo-OOVs toward open space and enhances separability.

### 3.5. Total Loss

Our method can be trained end-to-end by minimizing the following loss function:

$$
\mathcal{L} = \mathcal{L}_{align} + \mathcal{L}_{box} + \lambda \mathcal{L}_{\mathrm{LD\text{-}prior}}, \tag{15}
$$

where $\mathcal{L}_{align}$ represents the similarity-based cross-entropy classification loss, $\mathcal{L}_{box}$ represents the sum of the IoU loss and L1 regression loss of the prediction box. $\lambda$ is the weighting coefficient of the OOV objective function $\mathcal{L}_{\mathrm{LD\text{-}prior}}$.

*Table 1.* Zero-shot out-of-vocabulary object detection: comparisons on OOV-VOC and OOV-COCO. Best results are shown in **bold**, and second-best results are underlined. "↓" indicates that a smaller value is better, while "↑" indicates that a greater value is better.

| OOV Datasets | Methods | mAP$_{IV}$ / mAP$_{OOV}$ ↑ | mAP$_{Seen}$ / mAP$_{Unseen}$ ↑ | R$_{OOV}$ / AR$_{OOV}$ ↑ | WI / AOSE ↓ |
|---|---|---|---|---|---|
| **(a) OOV-VOC** | Max Energy (Liu et al., 2020) | 56.04 / 4.09 | 78.90 / 10.32 | 43.58 / 23.93 | 4.30 / 2381 |
| | MSP (Sun et al., 2021) | 55.51 / 5.50 | 78.34 / 9.84 | 47.71 / 24.38 | 4.83 / 2651 |
| | PROSER (Zhou et al., 2021) | 55.63 / 5.99 | 78.55 / 9.79 | 35.85 / 18.26 | 4.94 / 2648 |
| | OpenDet (Han et al., 2022) | 55.44 / 2.43 | 77.61 / 2.43 | 46.61 / 23.77 | 4.42 / 2439 |
| | GAIA (Chen et al., 2023) | 52.23 / 4.86 | 73.13 / 10.36 | 45.56 / 23.65 | 4.82 / 2260 |
| | EDL (Bao et al., 2021) | 57.90 / 2.02 | 79.13 / 15.45 | 40.83 / 20.76 | 4.84 / 2702 |
| | FOODv1 (Su et al., 2024) | 57.56 / 1.20 | 79.05 / 13.39 | 42.38 / 18.62 | 4.31 / 2727 |
| | FOODv2 (Su et al., 2023) | 56.95 / 0.54 | 79.01 / 12.82 | 44.06 / 22.21 | 4.78 / 2641 |
| | RegionCLIP (Zhong et al., 2022) | 53.70 / 5.60 | 76.19 / 8.74 | 45.08 / 23.17 | 4.96 / 2567 |
| | CoOp (Zhou et al., 2022a) | 52.24 / 7.95 | 76.55 / 3.64 | 47.43 / 23.47 | 4.64 / 2753 |
| | CED-FOOD (Wu et al., 2025) | 53.92 / 3.99 | 73.17 / 15.41 | 44.21 / 21.39 | 4.70 / 2228 |
| | APLGOS (Zhang et al., 2025b) | 42.62 / 3.39 | 73.02 / 10.82 | 47.53 / 23.84 | 4.80 / 2568 |
| | **Our OOVDet** | **58.70/10.65** | **79.30/21.40** | **56.42/28.12** | **3.95/2142** |
| **(b) OOV-COCO** | Max Energy (Liu et al., 2020) | 27.62 / 3.00 | 48.02 / 7.23 | 21.56 / 10.21 | 3.70 / 4278 |
| | MSP (Sun et al., 2021) | 27.51 / 1.79 | 45.85 / 9.17 | 18.92 / 7.38 | 3.83 / 4773 |
| | PROSER (Zhou et al., 2021) | 27.88 / 2.86 | 46.03 / 9.72 | 18.30 / 7.27 | 3.85 / 4783 |
| | OpenDet (Han et al., 2022) | 27.54 / 5.24 | 46.51 / 8.57 | 21.98 / 10.33 | 3.60 / 5228 |
| | GAIA (Chen et al., 2023) | 29.97 / 1.46 | 47.49 / 12.45 | 16.14 / 7.81 | 3.57 / 5148 |
| | EDL (Bao et al., 2021) | 26.59 / 5.22 | 47.17 / 6.00 | 20.45 / 9.60 | 3.61 / 4301 |
| | FOODv1 (Su et al., 2024) | 32.63 / 4.01 | 50.62 / 14.64 | 21.55 / 10.54 | 3.29 / 5194 |
| | FOODv2 (Su et al., 2023) | 32.03 / 3.59 | 50.04 / 13.67 | 22.14 / 10.93 | 3.18 / 4878 |
| | RegionCLIP (Zhong et al., 2022) | 25.70 / 4.59 | 44.74 / 6.66 | 16.82 / 7.33 | 3.34 / 8252 |
| | CoOp (Zhou et al., 2022a) | 31.36 / 4.26 | 48.77 / 15.31 | 22.83 / 10.94 | 3.95 / 5236 |
| | CED-FOOD (Wu et al., 2025) | 27.08 / 5.23 | 45.40 / 10.74 | 20.20 / 9.97 | 3.62 / 5596 |
| | APLGOS (Zhang et al., 2025b) | 31.60 / 4.08 | 50.01 / 13.19 | 22.42 / 10.69 | 3.32 / 5421 |
| | **Our OOVDet** | **33.19/10.09** | **51.15/15.89** | **27.60/12.82** | **3.12/4222** |

## 4. Experiments

### 4.1. Experimental Detail

**Datasets.** We evaluate ZS-OOVD on six datasets. As illustrated in Fig. 3, OOV-VOC (Mark et al., 2010; Su et al., 2023; Wu et al., 2025) splits PASCAL VOC into 10 seen classes ($C_{Seen}$), 5 unseen classes ($C_{Unseen}$), and 5 OOV classes ($C_{OOV}$). OOV-COCO (Lin et al., 2015; Su et al., 2023; Wu et al., 2025) further scales this setting to 20 VOC seen classes, 20 COCO unseen classes, and 40 COCO OOV classes for quantitative evaluation. In addition, we assess real-world generalization via qualitative OOV detection visualization on ObstacleTrack (Chan et al., 2021), ADE-OoD (Galesso et al., 2024), MVTec-Anomaly (Bergmann et al., 2019), and RoadAnomaly (Lis et al., 2020).

**Evaluation Metrics.** The **mean average precision (mAP)** is chosen to evaluate the performance of IV classes, including mAP$_{IV}$, mAP$_{OOV}$, mAP$_{Seen}$, and mAP$_{Unseen}$. For the out-of-vocabulary evaluation, the **recall (R$_{OOV}$)** and **average recall (AR$_{OOV}$)** is reported. Furthermore, we report **Wilderness Impact (WI)** under a recall level of 0.8 to measure the degree of OOV objects misclassified to IV ones: $WI = (\frac{P_{\mathcal{IV}}}{P_{\mathcal{IV} \cup \mathcal{OOV}}} - 1) \times 100$, where $P_{\mathcal{IV}}$ and $P_{\mathcal{IV} \cup \mathcal{OOV}}$ denote the precision of close-set and open-set classes, respectively. Following (Han et al., 2022), we report WI under a recall level of 0.8. In addition, we also use **the Absolute Open Set Error (AOSE)** (Miller et al., 2018) to measure

the total number of OOV objects misclassified as IV.

**Implementation Details.** We build upon RegionCLIP using ResNet50, an ImageNet-pretrained offline RPN, and a CC3M-pretrained transformer text encoder (Zhong et al., 2022). Unlike CoOp-style methods (Zhou et al., 2022a), we freeze both the prompt learner and text encoder, and precompute IV text embeddings as fixed semantic prototypes, allowing the model to focus on visual–semantic alignment and OOV boundary modeling. Training is performed on four NVIDIA RTX 3090 GPUs with batch size 4, using SGD (base learning rate is $5 \times 10^{-4}$), decayed once at 80k iterations, for 120k iterations without warm-up.

### 4.2. Main Results

**Experiments on OOV-VOC.** Table 1(a) shows that OOVDet outperforms state-of-the-art zero-shot OOV detectors on OOV-related metrics, achieving +2.70% mAP$_{OOV}$, +8.71% R$_{OOV}$, +3.74% AR$_{OOV}$ and reducing WI by 0.35 and AOSE by 86. This indicates stronger OOV recall with fewer IV false positives. OOVDet also maintains competitive IV performance, with mAP$_{Unseen}$=21.40% (up 5.95% over EDL (Bao et al., 2021)) and mAP$_{Seen}$=79.30%, demonstrating that OOV modeling does not compromise IV detection. Overall, OOVDet achieves a better balance between OOV sensitivity and IV stability.

**Experiments on OOV-COCO.** As shown in Table

*Table 2.* Ablation study of OOV prompt synthesis on OOV-VOC. We compare different prompt strategies and masking ratios under different backbones. Our ChatGPT-standardized Q&A prompts consistently outperform vanilla and handcrafted prompt variations in both IV and OOV metrics. Results demonstrate that class-conditional masked Gaussian perturbation $\vec{\varepsilon}$ effectively synthesizes informative OOV prompts, improving unseen-IV generalization and OOV rejection.

| Prompt Strategy | Backbone + Text emb. dim | mAP$_{IV}$ / mAP$_{OOV}$ ↑ | mAP$_{Seen}$ / mAP$_{Unseen}$ ↑ | R$_{OOV}$ / AR$_{OOV}$ ↑ | WI / AOSE ↓ |
|---|---|---|---|---|---|
| "a region of a" + <cls> | | 58.21 / 3.58 | 78.74 / 17.16 | 54.13 / 26.98 | 4.44 / 2456 |
| Common prompt variations (Zhong et al., 2022) | RN50+dim=1024 | 58.29 / 10.15 | 78.50 / 17.86 | 56.15 / 28.11 | 4.42 / 2450 |
| ChatGPT standardized Q&A pairs (Ours) | | **58.70 / 10.65** | **79.30 / 21.40** | **56.42 / 28.12** | **3.95 / 2142** |
| "a region of a" + <cls> | | 62.21 / 10.54 | 82.67 / 21.31 | 71.05 / 39.21 | 2.86 / 1669 |
| Common prompt variations (Zhong et al., 2022) | RN50x4+dim=640 | 61.80 / 11.17 | 82.44 / 25.34 | 73.57 / 37.29 | 2.93 / 1761 |
| ChatGPT standardized Q&A pairs (Ours) | | **63.63 / 11.20** | **83.03 / 26.01** | **74.53 / 39.75** | **2.79 / 1695** |
| mask ratio=0.25 | | 58.17 / 9.67 | 78.79 / 16.95 | **70.66 / 35.07** | 4.01 / 2487 |
| **mask ratio=0.50** | RN50+dim=1024 | **58.70 / 10.65** | **79.30 / 21.40** | 56.42 / 28.12 | **3.95 / 2142** |
| mask ratio=0.75 | | 58.68 / 7.06 | 78.19 / 19.66 | 55.76 / 27.94 | 4.45 / 2406 |
| mask ratio=1.00 | | 58.14 / 5.58 | 78.52 / 21.38 | 44.75 / 23.18 | 4.39 / 2642 |

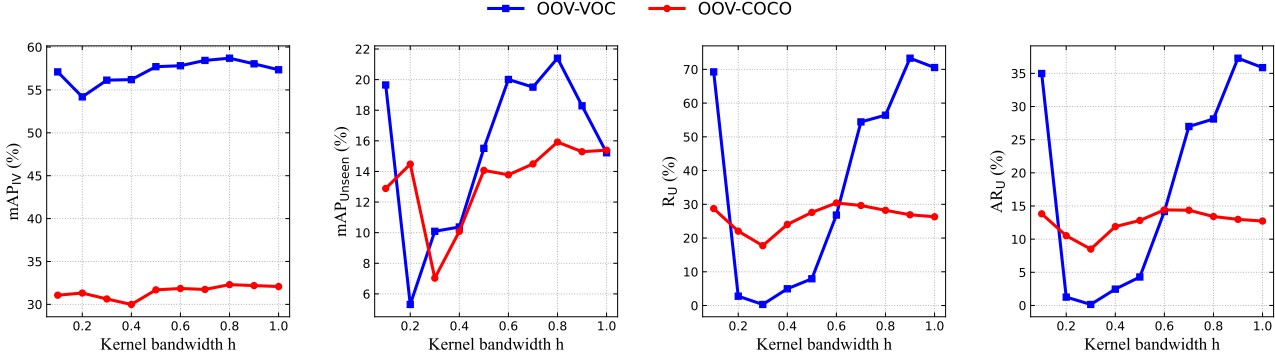

*Figure 4.* Effect of kernel bandwidth $h$ on IV, unseen-IV, and OOV detection performance. We analyze the kernel bandwidth $h$ in the Low-Density Prior Constraint on OOV-VOC and OOV-COCO, showing that $h$ strongly affects both IV and OOV performance. A moderate bandwidth ($h = 0.8$) provides the best trade-off, maximizing IV accuracy while preserving unseen-IV generalization and stable OOV detection. The blue and red lines correspond to OOV-VOC and OOV-COCO evaluations, respectively.

*Table 3.* Ablation study on pseudo-OOV image construction.

| Top-k | fg:bg | mAP$_{IV}$↑ | mAP$_{Unseen}$↑ | R$_U$↑ | AR$_U$↑ |
|---|---|---|---|---|---|
| 1 | 1:1 | 57.77 | 21.29 | 42.10 | 21.77 |
| **3** | **1:1** | **58.70** | **21.40** | 56.42 | 28.12 |
| 5 | 1:1 | 58.64 | 16.85 | **71.70** | **36.11** |
| 10 | 1:1 | 55.46 | 19.56 | 53.58 | 27.84 |
| 3 | 1:2 | 57.49 | 15.22 | 71.44 | 36.06 |
| 3 | 1:3 | 57.15 | 20.43 | 57.96 | 29.10 |

1(b), OOVDet achieves the best mAP$_{OOV}$=10.09%, with R$_{OOV}$=27.60% and AR$_{OOV}$=12.82%, demonstrating strong localization and recall for diverse OOV instances. In addition, OOVDet yields the lowest WI=3.12 and AOSE=4222 among all compared methods, indicating that the proposed approach significantly reduces confusion between in-vocabulary and out-of-vocabulary categories and suppresses false unknown predictions. Despite the increased difficulty of this benchmark, OOVDet maintains strong IV generalization, achieving mAP$_{Seen}$=51.15% and mAP$_{Unseen}$=15.89%, which confirms that the pseudo-OOV modeling strategy scales effectively to large OOV spaces without compromis-

ing recognition of known categories.

### 4.3. Ablation Study

**Ablation on OOV Prompt Synthesis.** Table 2 shows that under RN50, our ChatGPT-standardized Q&A prompts achieve 58.70% mAP$_{IV}$ and 21.40% mAP$_{Unseen}$, improving AR$_{OOV}$ (26.98%→28.12%) and reducing WI (4.44→3.95) compared to the vanilla template. Similar gains are observed with RN50x4 (63.63% / 26.01%), indicating good robustness across backbones. A mask ratio of 0.50 yields the best trade-off, validating that our prompt synthesis with moderate perturbation effectively generates informative OOV prompts in low-density regions.

**Ablation on Pseudo-OOV Image.** Table 3 shows that increasing Top-$k$ from 1 to 3 improves mAP$_{IV}$ (57.77%→58.70%), mAP$_{Unseen}$ (21.29%→21.40%) and OOV recall $R_U$ (42.10%→56.42%), while larger Top-$k$ (5 or 10) increases recall but degrades unseen performance due to noisy gradients. Increasing background proportion also boosts $R_U$ but harms mAP$_{Unseen}$. The best balance is

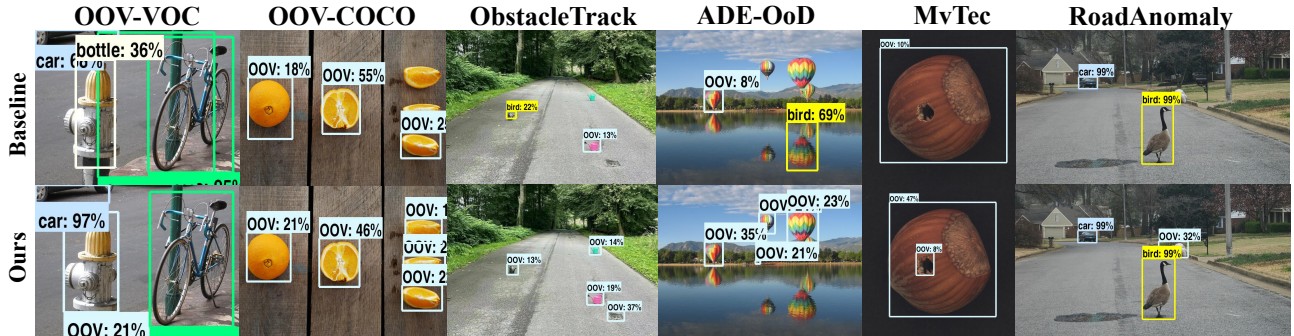

*Figure 5.* The visualized results on OOV-VOC,OOV-COCO and diverse real-world datasets.

*Table 4.* Effect of kernel bandwidth $h$ on OOV-VOC.

| $h$ | mAP$_{IV}$↑ | mAP$_{Unseen}$↑ | R$_U$↑ | AR$_U$↑ |
|---|---|---|---|---|
| 0.1 | 57.10 | 19.65 | 69.25 | 34.98 |
| 0.2 | 54.19 | 5.31 | 2.77 | 1.26 |
| 0.3 | 56.14 | 10.09 | 0.32 | 0.15 |
| 0.4 | 56.20 | 10.37 | 4.96 | 2.46 |
| 0.5 | 57.71 | 15.51 | 7.93 | 4.29 |
| 0.6 | 57.82 | 20.01 | 26.82 | 14.17 |
| 0.7 | 58.44 | 19.51 | 54.42 | 26.99 |
| **0.8** | **58.70** | **21.40** | 56.42 | 28.12 |
| 0.9 | 58.04 | 18.29 | **73.31** | **37.29** |
| 1.0 | 57.35 | 15.22 | 70.54 | 35.87 |

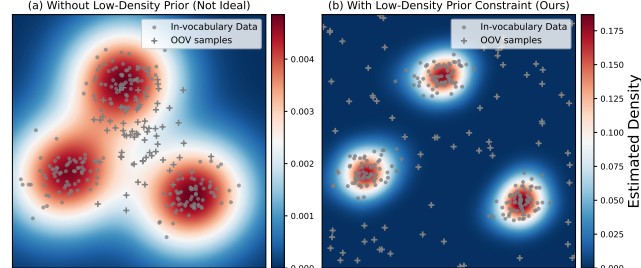

*Figure 6.* Visualization of the proposed Low-Density Prior Constraint in the feature space.

achieved at Top-$k = 3$ and fg:bg=1:1, validating DGA's effectiveness.

**Ablation on OOV Optimization.** As shown in Fig. 4, A too small h produces noisy and fragmented densities, leading to unstable OOV separation, whereas a too large h oversmooths the class structure and weakens discriminability. As a result, h implicitly determines the decision boundary between IV and OOV samples, making the overall performance sensitive to its value. This effect is consistent across datasets: on OOV-VOC (Table 4), $h = 0.8$ achieves the best mAP$_{IV}$ (58.70%) while maintaining strong unseen-IV generalization (mAP$_{Unseen}$=21.40%) and competitive OOV detection (AR$_U$=28.12%). A similar trend holds on OOV-COCO. Thus, $h = 0.8$ is adopted in all experiments.

**Effect of the Low-Density Prior Constraint.** Fig. 6 visualizes the Low-Density Prior Constraint. Without it, IV density is diffuse and OOV samples (gray "+") lie near IV clusters, causing entangled decision regions. With the constraint, IV features form compact high-density modes and OOV samples are pushed into low-density valleys, yielding clearer boundaries. This aligns with our design, where high-uncertainty OOV-related samples in high IV density regions are penalized, improving IV compactness, unseen-IV generalization, and OOV rejection.

### 4.4. Visualization of OOV Detection Results

Fig. 5 shows that on OOV-VOC and OOV-COCO, the baseline often misclassifies OOV objects as semantically similar IV categories with high confidence, whereas our method assigns higher OOV confidence and suppresses incorrect IV predictions, yielding clearer IV-OOV separation. On more complex benchmarks with severe domain shifts and complex visual patterns, including ObstacleTrack, ADE-OoD, MVTec Anomaly, and RoadAnomaly, the baseline frequently produces unstable, erroneous high-confidence IV detections on OOV regions. In contrast, our approach remains robust by reliably identifying unknown and anomalous regions as OOV while maintaining stable IV detection, reflecting more trustworthy prediction confidence in open-world scenarios. Additional visualizations and ablations are provided in Appendix.

## 5. Conclusion

This paper studies zero-shot out-of-vocabulary detection (ZS-OOVD), an underexplored yet critical problem in open-world object detection, which requires reliable in-vocabulary recognition while rejecting undefined objects without training samples or semantic prompts. We propose OOVDet, a unified framework integrating OPS, DGA, and LPC to explicitly model the discrepancy between in-vocabulary and out-of-vocabulary samples, enabling robust IV-OOV discrimination. Extensive experiments across multiple bench-

marks validate the effectiveness of the proposed approach. Overall, this work takes a practical step toward reliable open-world object detection under zero-shot constraints and highlights the importance of OOV-aware representation and decision boundary modeling.

## Acknowledgements

This work was supported in part by the National Key Research and Development Program of China under Grant 2024YFB3310900, in part by the Central Government Guides Local Science and Technology Development Fund under Projects 246Z4306G and 246Z1602G, in part by the Shijiazhuang Science and Technology Cooperation Project under Grant SJZZXC24009, in part by the Tianjin Municipal Education Commission Scientific Research Project under Grant 2024KJ151, in part by the Hebei Province Yanzhao Huangjintai Talent Gathering Program Backbone Personnel (Postdoctoral Platform) Project under Grant B2025005031, and in part by the Technical Service for Image Quality Inspection of Photovoltaic Modules Project (Contract No. HI2445) funded by Hebei Fengxing Power Sales Co., Ltd.

## Impact Statement

This paper presents work whose goal is to advance the field of Machine Learning. There are many potential societal consequences of our work, none which we feel must be specifically highlighted here.

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
