# OpenReview forum: "OOVDet: Low-Density Prior Learning for Zero-Shot Out-of-Vocabulary Object Detection"
_ICML.cc/2026/Conference — ICML 2026 regular_

### Official Review · Reviewer_39Zx · 2026-03-08

**Soundness:** 3
**Presentation:** 3
**Significance:** 3
**Originality:** 2
**Overall Recommendation:** 4
**Confidence:** 2

**Summary:**

This paper proposes OOVDet, a framework for zero-shot out-of-vocabulary object detection that synthesizes OOV prompts, mines pseudo-OOV samples, and introduces a low-density prior constraint to improve the separation between in-vocabulary and undefined objects.

**Compliance With Llm Reviewing Policy:**

Affirmed.

**Final Justification:**

Thank authors for addressing my concern. I will keep my original score.

**Key Questions For Authors:**

The paper introduces ZS-OOVD as a new task formulation. It would be helpful if the authors could further clarify how this setting differs from related problems such as open-world object detection or open-set detection.

The DGA module selects high-uncertainty proposals as pseudo-OOV samples during training. It would be interesting if the authors could provide additional discussion or analysis on the characteristics of these pseudo-OOV samples. For example, understanding whether these proposals tend to correspond to unknown objects or difficult in-vocabulary examples could provide further insight into the training dynamics.

**Limitations:**

No，The paper does not explicitly discuss the limitations of the proposed method or potential societal impacts. It would be helpful if the authors could briefly discuss possible limitations, such as the reliance on pseudo-OOV samples during training and the potential sensitivity of the method to the quality of prompt embeddings generated in the OPS module.

**Strengths And Weaknesses:**

The proposed framework is technically reasonable and supported by experiments and ablation studies on OOV-VOC and OOV-COCO. The paper is generally well organized and clearly structured. The motivation for the ZS-OOVD problem is clearly introduced, and the overall framework is illustrated in an informative figure that helps readers understand the interaction between the three proposed components. The experimental section is reasonably detailed, including implementation details and multiple ablation studies. Overall, the narrative is easy to follow and the method pipeline is clearly explained.

The proposed approach builds upon several existing ideas from open-set detection and uncertainty-based sample mining. While the integration is effective, the degree of algorithmic novelty is somewhat limited.
The distinction between the proposed ZS-OOVD setting and related tasks such as open-world or open-set object detection could be discussed more clearly to better position the contribution.

---

> ### Author Rebuttal · Authors · 2026-03-31
>
> Paper 11408:
>
> Weaknesses
>
> Q1: The proposed approach builds upon several existing ideas from open-set detection and uncertainty-based sample mining. **(1) While the integration is effective, the degree of algorithmic novelty is somewhat limited**. (2) **The distinction between the proposed ZS-OOVD setting and related tasks** such as open-world or open-set object detection could be discussed more clearly to better position the contribution.
>
> A1: (1) Regarding our proposed algorithm, the novelty resides in the language-space modeling of pseudo-OOV prompts, which, to the best of our knowledge, has not been investigated in previous literature. We further characterize channel-wise gradient responses using a generalized Dirichlet distribution, and quantify proposal uncertainty $U_G$ via negative log-likelihood (see Section 3.3). In addition, we propose to model pseudo-OOV samples via Gaussian kernel density estimation and explicitly push them into low-density regions through a novel $L_{LD-prior}$ loss (see Eq. 12-14), leading to clearer IV–OOV boundaries.
>
> (2) The distinction between the proposed ZS-OOVD setting and related tasks such as open-world or open-set object detection lies primarily in a fresh **language-driven perspective** for task formulation **(see Fig. 1 and Fig. 3)**. Prior tasks (e.g., OWOD, OSOD) are image-centric unknown rejection methods, focusing on recognizing known classes while rejecting unknown ones, **whereas out-of-vocabulary detection (OOVD) introduces a novel language-driven viewpoint, leveraging whether an object can be described in language as the criterion to delineate defined and undefined categories**. Thus, OOVD can be seamlessly integrated with zero-shot learning (our ZS-OOVD), enabling more versatile and generalizable object detection. As suggested, we would discuss more clearly to better position the contribution in the revision.
>
> Key Questions
>
> KQ1: The paper introduces ZS-OOVD as a new task formulation. It would be helpful if the authors could further clarify **how this setting differs from related problems such as open-world object detection or open-set detection**.
>
> A2: The relevant explanation can **refer to A1 (2)**.
>
> KQ2: The DGA module selects high-uncertainty proposals as pseudo-OOV samples during training. It would be interesting **if the authors could provide additional discussion or analysis on the characteristics of these pseudo-OOV samples**. For example, understanding whether these proposals tend to correspond to unknown objects or difficult in-vocabulary examples could provide further insight into the training dynamics.
>
> A3: As suggested, additional discussion or analysis will be provided to better characterize the pseudo-OOV samples selected by the DGA module. In particular, the discussion will focus on understanding whether these high-uncertainty proposals correspond to unknown objects, difficult IV examples, or noisy background regions. Further quantitative evaluation, such as a bar chart illustrating the proportions of unknown OOV objects and difficult in-vocabulary (IV) examples, will be included, along with estimates of the proportions of each sample type in the validation set, thereby offering a clearer view of mining precision and its role in training dynamics.
>
> Limitations
>
> L1: **The paper does not explicitly discuss the limitations of the proposed method or potential societal impacts.** It would be helpful if the authors could briefly discuss possible limitations, such as the reliance on pseudo-OOV samples during training and the potential sensitivity of the method to the quality of prompt embeddings generated in the OPS module.
>
> A4: Thank you for your suggestion. The reliance on pseudo-OOV samples introduces a potential limitation, as these samples may not perfectly align with real-world OOV distributions and could introduce biases during training. In addition, the method may be sensitive to the quality of prompt embeddings generated in the OPS module, since suboptimal semantic representations can affect the accuracy of OOV detection and the delineation of decision boundaries. **These limitations will be explicitly discussed and further analyzed in the supplementary material**.

---

> > ### Author Rebuttal · Reviewer_39Zx · 2026-04-02
> >
> > Thank you for mentioning in the rebuttal that you plan to conduct further experiments on the second key question. Could you please include some experimental results to support this point?

---

> > > ### Author Response · Authors · 2026-04-04
> > >
> > > Paper 11408:
> > >
> > > KQ2: The DGA module selects high-uncertainty proposals as pseudo-OOV samples during training. It would be interesting if the authors could provide additional discussion or analysis on the characteristics of these pseudo-OOV samples. For example, understanding **whether these proposals tend to correspond to unknown objects or difficult in-vocabulary examples** could provide further insight into the training dynamics.
> > >
> > > Q1: Thank you for mentioning in the rebuttal that you plan to conduct further experiments on the second key question. **Could you please include some experimental results to support this point?**
> > >
> > > A1: Thanks for your comment. During the training of OOVDet on OOV-VOC, we performed pseudo-OOV image sample mining 100 times, with a sampling interval of 1200 iterations. At each sampling step, 6 pseudo-OOV samples were selected (top-k = 3, with a foreground-to-background ratio of 1:1, see Table 3), including 3 from the foreground and 3 from the background, resulting in a total of 600 pseudo-OOV samples. We further analyze the composition of these samples by categorizing them into **(1) Unknown objects (True OOV), (2) Hard IV examples (Seen), (3) Hard IV examples (Unseen),** and **(4) True background**, as summarized in the table.
> > > | Category                            | Foreground | Background | Total |
> > > |-------------------------------------|------------|------------|-------|
> > > | Unknown objects (True OOV)           | 0          | 33         | 33    |
> > > | Hard IV examples (Seen)         | 300        | 0          | 300   |
> > > | Hard IV examples (Unseen)       | 0          | 58         | 58    |
> > > | True background                     | 0          | 209        | 209   |
> > > | **Total**             | **300**    | **300**    | **600** |
> > >
> > > Specifically, all foreground samples (300/300) are drawn from known IV categories, and thus no true OOV instances are present in the foreground. In contrast, true OOV samples account for only 33/600 (5.5%) and are predominantly located in background regions, along with 58 hard unseen IV samples and 209 true background samples. Overall, this distribution reveals a clear and systematic discrepancy between the mined pseudo-OOV samples and the true OOV distribution, indicating that current pseudo-OOV mining primarily captures boundary-hard IV samples rather than semantically novel OOV instances.

---

### Official Review · Reviewer_1GKe · 2026-03-10

**Soundness:** 3
**Presentation:** 4
**Significance:** 3
**Originality:** 3
**Overall Recommendation:** 4
**Confidence:** 4

**Summary:**

This paper studies zero-shot out-of-vocabulary detection (ZS-OOVD), a setting where an object detector must recognize objects belonging to predefined categories (including zero-shot unseen classes) while rejecting objects that fall outside the vocabulary. The problem extends traditional zero-shot object detection (ZSOD) by explicitly requiring reliable rejection of undefined categories.

The authors aim to consider a core challenge in open-world perception: enabling models to recognize known objects while simultaneously identifying truly unknown objects in a zero-shot setting without explicit supervision for those unknown classes.

To address this problem, the paper proposes OOVDet, a framework composed of three components:

Out-of-Vocabulary Prompt Synthesis (OPS)
Synthesizes OOV prompt embeddings by sampling from the low-likelihood regions of class-conditional Gaussian distributions modeled from in-vocabulary (IV) prompt embeddings.

Dirichlet-based Gradient Attribution (DGA)
Uses attribution gradients to estimate uncertainty via a Dirichlet distribution and mines high-uncertainty region proposals as pseudo-OOV samples.

Low-Density Prior Constraint (LPC)
Uses Gaussian kernel density estimation to enforce that pseudo-OOV samples lie in low-density regions of the feature space between IV clusters.

The framework is implemented on top of a RegionCLIP-style detector and evaluated on OOV-VOC and OOV-COCO benchmarks, showing improvements in OOV recall, open-set error, and OOV detection performance while maintaining competitive IV accuracy.

Overall, the study's broad topic concerns open-world object detection and robust handling of unknown objects in vision-language detection systems.

**Compliance With Llm Reviewing Policy:**

Affirmed.

**Key Questions For Authors:**

- Computational overhead

What is the additional training and inference cost introduced by the Dirichlet-based gradient attribution module compared to simpler uncertainty estimation methods?

- Prompt sensitivity

How sensitive is the OPS module to the prompt generation process? Would the method perform similarly with simple prompt templates instead of LLM-generated ones?

- Distributional assumption validation

Can the authors provide empirical analysis demonstrating that OOV samples indeed lie in low-density regions of the IV feature space?

- Kernel bandwidth sensitivity

How sensitive is the LPC module to the Gaussian kernel bandwidth parameter $h$?

- Scalability

How does the framework scale to large vocabularies with hundreds or thousands of predefined classes?

**Limitations:**

Partially.

The paper discusses the difficulty of modeling unknown classes without supervision but does not fully address:

the limitations of the Gaussian distribution assumption

sensitivity to prompt generation

possible failure cases where OOV objects lie close to IV clusters.

Adding a clearer discussion of these limitations would improve the paper.

**Strengths And Weaknesses:**

Strengths

1. Well-motivated and practically relevant problem

The paper tackles a realistic and underexplored problem: distinguishing objects that fall outside a predefined vocabulary during zero-shot inference.

While zero-shot and open-vocabulary detection have been widely studied, explicit out-of-vocabulary rejection without training examples remains relatively underdeveloped.

The ZS-OOVD formulation provides a useful conceptual distinction between:

- unseen but predefined categories

- truly undefined categories.

2. Conceptually coherent framework

The proposed method integrates three complementary mechanisms that address different aspects of the OOV problem:

- OPS models semantic uncertainty in prompt space

- DGA identifies uncertain visual samples

- LPC enforces density separation in representation space

This unified design provides a consistent approach to modeling the boundary between IV and OOV classes.

3. Creative use of uncertainty modeling

The Dirichlet-based gradient attribution mechanism for pseudo-OOV mining is an interesting departure from standard approaches such as:

- maximum softmax probability

- energy-based OOD detection

- entropy-based uncertainty.

Modeling gradient distributions as Dirichlet evidence introduces a novel perspective on uncertainty estimation in detection models.

4. Solid empirical results

The proposed approach demonstrates consistent improvements on both evaluation datasets.

For example:

- OOV recall (ROOV) improves by up to 8.7%

- Absolute open-set error (AOSE) decreases significantly

- IV detection performance remains competitive.

These results suggest the framework improves OOV detection without significantly sacrificing IV accuracy.

5. Detailed ablation studies

The paper includes thorough ablation analyses covering:

- prompt synthesis strategies

- masking ratios for OPS

- pseudo-OOV mining parameters

- kernel bandwidth for density estimation.

These experiments provide useful insights into the behavior of the proposed modules.

Weaknesses
1. Limited conceptual novelty in core assumptions

Although the framework is well engineered, several components build upon existing open-set detection principles, including:

- low-density separation assumptions

- pseudo-unknown sample mining

- density-based regularization.

The primary novelty lies in combining these ideas for zero-shot OOV detection, rather than introducing fundamentally new modeling principles.

2. Strong assumptions about feature distributions

The method assumes that OOV samples lie in low-density regions between IV clusters in the latent space.

However, in real-world scenarios:

- OOV samples may form dense clusters

- unseen classes may lie near IV classes

- embedding distributions may not be Gaussian.

The paper would benefit from empirical evidence validating this assumption.

3. Reliance on LLM-generated prompts

The OPS module depends on prompt variations generated by ChatGPT to construct the IV prompt distribution.

This introduces potential concerns regarding:

- reproducibility

- sensitivity to prompt diversity

- dependence on external models.

A comparison with simpler prompt templates would clarify the necessity of this design choice.

4. Computational overhead

The framework introduces several additional operations compared to standard detectors:

- gradient attribution modeling

- Dirichlet parameter estimation

- kernel density estimation.

Although approximations are proposed, the paper does not quantify training or inference overhead.

5. Benchmark novelty could be clearer

While the paper introduces the ZS-OOVD task and evaluation protocol, the benchmark appears to be largely based on reconfigured versions of existing datasets (VOC and COCO).

Clarifying what aspects of the benchmark are truly new would strengthen the contribution.

---

> ### Author Rebuttal · Authors · 2026-03-31
>
> Paper 11408:
>
> Weakness
>
> Q1: Several components build upon existing open-set detection principles, including **(1) low-density separation assumptions, (2) pseudo-unknown sample mining, (3) density-based regularization**. The primary novelty lies in combining these ideas for zero-shot OOV detection, rather than introducing fundamentally new modeling principles.
>
> A1: Regarding the above three components: (1) The novelty lies in language-space modeling of pseudo-OOV prompts, which, to the best of our knowledge, has not been explored in prior work. (2) We propose to model channel-wise gradient responses using a generalized Dirichlet distribution, and quantify proposal uncertainty via the negative log-likelihood (see Section 3.3). (3) We model pseudo-OOV samples via Gaussian kernel density estimation and explicitly push them into low-density regions through a novel $L_{LD-prior}$ loss (see Eq. 12-14), leading to clearer IV–OOV boundaries.
>
> Q2: The method assumes OOV samples lie in low-density regions between IV clusters. In real-world scenarios **(1) OOV samples may form dense clusters, (2) unseen classes may lie near IV classes, (3) embedding distributions may not be Gaussian**. The paper would benefit from empirical evidence validating this assumption.
>
> A2: (1) The objective is to ensure clear separation between OOV and IV classes, **thus clustered OOV samples do not compromise the IV–OOV decision boundary.** (2) Unseen classes may lie near IV classes, **therefore, we mine them via uncertainty estimation and leverage these ambiguous samples to refine the IV–OOV decision boundary.** (3) The Gaussian assumption is adopted merely as a practical approximation. **Alternative distributional assumptions (e.g., vMF, t-distribution)** will be analyzed to demonstrate the effectiveness of our method.
>
> Q3: The OPS module depends on prompt variations generated by ChatGPT to construct the IV prompt distribution. This introduces potential concerns regarding: **(1) reproducibility, (2) sensitivity to prompt diversity, dependence on external models**. A comparison with simpler prompt templates would clarify the necessity of this design choice.
>
> A3: (1) For **reproducibility**, we will provide the concept_embeds.pth file in our codebase, which contains embeddings generated by the text encoder using prompt variations created by GPT.
>
> (2) Table 2 has shown that our prompts outperform vanilla and handcrafted templates across backbones, demonstrating robustness to prompt variations. **External LLMs** are adopted, as they can improve performance without disrupting end-to-end training (see Table 2 line 3).
>
> Q4: The framework introduces several additional operations **compared to standard detectors**, gradient attribution modeling, Dirichlet parameter estimation, kernel density estimation. Although approximations are proposed, the paper does not quantify training or inference overhead.
>
> A4: Compared to the baseline, our method incurs a slightly longer training time (11h20m vs. 9h49m), while maintaining comparable inference speed (15m30s vs. 15m24s on 4952 images) and a similar number of parameters (121.310M vs. 121.309M). This indicates that our improvements stem from algorithm-level modifications, without introducing additional parameters or increasing computational cost during inference.
>
> Q5: While the paper introduces the ZS-OOVD task and evaluation protocol, the benchmark appears to be largely based on reconfigured versions of existing datasets (VOC and COCO). **Clarifying what aspects of the benchmark are truly new would strengthen the contribution.**
>
> A5: Unlike VOC and COCO, which assume closed-set settings, our benchmark explicitly models open-world conditions via **a novel language-driven data partitioning (Seen/Unseen/OOV)** (see Fig. 3). Moreover, it goes beyond standard mAP by introducing OOV-specific metrics ($R_{OOV}$, $AR_{OOV}$, WI, AOSE) to directly evaluate OOV detection and misclassification risk.
>
> Key questions
>
> For the questions KQ1/KQ2/KQ3, please refer to A4/A3/A2, respectively.
>
> KQ4: How sensitive is the LPC module to the Gaussian kernel bandwidth parameter?
>
> A6: In Table 4, the **LPC kernel bandwidth h** is critical for Gaussian kernel density estimation. It directly controls **the smoothness of the estimated density and further shapes the low-density regions for OOV modeling**. A too small h causes noisy, fragmented densities and unstable OOV separation, while a too large h over-smooths class structure and weakens discriminability (see Fig. 4). **Consequently, h implicitly defines the decision boundary between IV and OOV samples**, making performance inherently sensitive to its value.
>
> KQ5: How does the framework scale to large vocabularies with hundreds or thousands of predefined classes?
>
> A7: The framework scales effectively to large vocabularies through its language-based design, requiring only updates to semantic prompt embeddings, with no need to restructure the core model.

---

> > ### Author Rebuttal · Reviewer_1GKe · 2026-04-05
> >
> > Thank you for the detailed rebuttal and additional analyses.
> >
> > My concerns have been partially addressed. In particular, the added quantitative evidence (e.g., likelihood comparisons between IV and strict real OOV categories, and comparisons with alternative distributions) provides useful empirical support for the low-density prior assumption. The additional analysis of pseudo-OOV sample composition is also appreciated.
> >
> > However, my core concern is not fully resolved. While the likelihood analysis demonstrates that OOV samples tend to have lower likelihood under IV distributions, it does not fully justify low-density regions as the most appropriate inductive bias for modeling OOV semantics. Additionally, the pseudo-OOV analysis shows that many mined samples correspond to hard IV or background instances, suggesting that improvements may partly arise from boundary refinement rather than explicit OOV modeling.
> >
> > That said, I acknowledge that the rebuttal strengthens the empirical support and improves clarity. My remaining concerns are more fundamental and would likely require deeper analysis (e.g., visualization or broader validation) rather than short rebuttal additions.

---

> > > ### Author Response · Authors · 2026-04-07
> > >
> > > Paper 11408:
> > >
> > > Q: My concerns have been partially addressed. In particular, the added quantitative evidence (e.g., likelihood comparisons between IV and strict real OOV categories, and comparisons with alternative distributions) provides useful empirical support for the low-density prior assumption.
> > >
> > > The additional analysis of pseudo-OOV sample composition is also appreciated. However, my core concern is not fully resolved. While the likelihood analysis demonstrates that OOV samples tend to have lower likelihood under IV distributions, **(1) it does not fully justify low-density regions as the most appropriate inductive bias for modeling OOV semantics**. Additionally, (2) **the pseudo-OOV analysis shows that many mined samples correspond to hard IV or background instances, suggesting that improvements may partly arise from boundary refinement rather than explicit OOV modeling**.
> > > That said, I acknowledge that the rebuttal strengthens the empirical support and improves clarity. My remaining concerns are more fundamental and would likely require deeper analysis (e.g., visualization or broader validation) rather than short rebuttal additions.
> > >
> > > A: Thanks for your comment. (1) Low-likelihood region does not the most appropriate inductive bias for modeling OOV semantics. However, we emphasize that **modeling strict OOV semantics is inherently ill-posed** due to the absence of explicit language descriptions or labeled samples. Under this setting, it is natural to leverage the observable structure induced by IV distributions.
> > >
> > > Our method leverages this structure by modeling IV embeddings with class-conditional Gaussian distributions, where high-likelihood regions correspond to well-aligned, semantically defined concepts, while low-likelihood regions naturally capture areas not explained by IV semantics. Therefore, low-likelihood regions are not an arbitrary design choice, but a data-driven and tractable approximation of semantic “OOV” under limited supervision.
> > >
> > > (2) We further note that even though pseudo-OOV samples may include hard IV or background instances, the consistent performance gains in OOV detection  **(see page 6 Table 1)** suggest that the proposed inductive bias extends beyond boundary refinement and effectively captures unknown semantics in practice **(see page 14 Figure 9)**.
> > >
> > > These results provide direct empirical evidence of a consistent likelihood gap between IV and OOV samples, supporting low-likelihood regions as a practical and effective surrogate for modeling OOV semantics under ill-posed conditions. **For deeper analysis, t-SNE visualizations and broader validation will be included in the revision to further reinforce this conclusion.**

---

### Official Review · Reviewer_ex8A · 2026-03-12

**Soundness:** 3
**Presentation:** 3
**Significance:** 3
**Originality:** 3
**Overall Recommendation:** 4
**Confidence:** 4

**Summary:**

This paper studies the domain of zero-shot out-of-vocabulary detection (ZS-OOVD). In this setting, models must identify categories within a predefined vocabulary during testing while rejecting undefined objects not present in the vocabulary. To this end, the authors propose OOVDet, introducing three modules to the RegionCLIP baseline. These modules are designed at three distinct levels: text-level OOV prompt synthesis, image-level pseudo-OOV region mining, and loss function-level decision boundary optimisation. Experiments on OOV-VOC and OOV-COCO demonstrate that the method outperforms comparable approaches on OOV-related metrics while maintaining relatively competitive in-vocabulary detection performance.

**Compliance With Llm Reviewing Policy:**

Affirmed.

**Final Justification:**

The rebuttal adequately addresses my main concerns for rebuttal purposes. In particular, the authors clarify that a central contribution of the work lies in the proposed ZS-OOVD task formulation, and they further explain how the proposed method is designed to support this task setting. The response also provides additional clarification regarding the low-density assumption, the role of each component, and the sensitivity of the LPC bandwidth parameter. While some of these aspects could still be strengthened further in the final version, I consider my original concerns sufficiently addressed at this stage.

**Key Questions For Authors:**

1. The core hypothesis of this paper is that out-of-vocabulary (OOV) semantics and visual samples are more likely to reside in low-density regions of known-class feature spaces. Could the authors provide stronger evidence for this assumption, ideally through quantitative analysis and, if helpful, additional visualizations demonstrating that it is a reasonable and sufficiently general assumption?
2. While the three modules mentioned are all motivated by the same low-density hypothesis, the paper does not provide enough analysis to show that they form a genuinely synergistic design rather than a loose combination of OOV-oriented heuristics. Could the authors further clarify the contribution of each component and provide an analysis of their joint effect?

**Limitations:**

The paper does not sufficiently analyze the validity of the low-density assumption. Furthermore, the relatively low absolute performance on OOV-COCO suggests that important unresolved issues remain in the current framework, and these limitations should be discussed more explicitly.

**Strengths And Weaknesses:**

• STRENGTH
• The research question addressed in this paper holds practical significance. The requirement that the model explicitly rejects out-of-vocabulary objects presents a compelling point of departure and aligns more closely with open-world deployment scenarios.
• The method is logically structured, with three coordinated components operating at the text, image, and decision-boundary levels. The overall design is coherent and directly targets the central OOV detection problem discussed in the paper.
• The experimental results demonstrate that the algorithm does indeed deliver performance improvements.

• WEAKNESS
• Although the paper frames the problem as a joint ZS-OOVD task, the methodology is almost entirely centred on OOV modelling, with no direct novel mechanisms proposed for seen/unseen zero-shot recognition itself. Consequently, the contribution of this work appears more akin to enhancing OOV rejection within existing ZSOD frameworks than presenting a balanced and unified solution for the entire joint task. The paper requires clearer articulation of its core contribution: whether it lies in the novel task formulation, the OOV modelling mechanism, or a combination of both.
• Table 4 appears to indicate that LPC is highly sensitive to parameter selection, which may prove disadvantageous.

---

> ### Author Rebuttal · Authors · 2026-03-31
>
> Paper 11408:
>
> Weaknesses
>
> Q1: Although this work formulates the task as joint ZS-OOVD, **the methodology focuses predominantly on OOV modeling, with no novel dedicated mechanisms for standard zero-shot recognition of seen/unseen classes.** As such, its contribution is more aligned with improving OOV rejection within existing ZSOD frameworks, rather than providing a balanced, unified solution for the full joint task. The paper requires **clearer articulation of its core contribution**: whether it lies in the novel task formulation, the OOV modelling mechanism, or a combination of both.
>
> A1: **The core contribution lies in the novel task formulation**, which introduces a fresh **language-driven perspective**: it uses whether an object can be semantically described in language as the criterion to distinguish between defined and undefined classes (see page 1 lines 49-53). **This enables seamless integration of OOV detection with zero-shot learning, while maintaining competitive IV unseen (zero-shot) performance (+5.95% over the second in Table 1(a)).** To our knowledge, this formulation has not been explored in prior work.
> **Regarding the OOV modeling mechanism**, different from existing methods (e.g., OpenDet) that mine pseudo-samples from the visual domain, we propose to generate pseudo-OOV prompt embeddings from a class-conditional Gaussian space built upon IV language prompts (see Section 3.1 and 3.2), thus achieving effective **visual–language** alignment for zero-shot OOV detection.
>
> Q2: Table 4 appears to indicate that **LPC is highly sensitive to parameter selection**, which may prove disadvantageous.
>
> A2: In Table 4, the **LPC kernel bandwidth h** is critical for Gaussian kernel density estimation. It directly controls **the smoothness of the estimated density and further shapes the low-density regions for OOV modeling**. A too small h causes noisy, fragmented densities and unstable OOV separation, while a too large h over-smooths class structure and weakens discriminability (see Fig. 4). **Consequently, h implicitly defines the decision boundary between IV and OOV samples**, making performance inherently sensitive to its value.
>
> Key Questions
>
> KQ1: The core hypothesis of this paper is that OOV semantics and visual samples are more likely to **reside in low-density regions** of known-class feature spaces. **Could the authors provide stronger evidence for this assumption**, ideally through quantitative analysis and, if helpful, additional visualizations demonstrating that it is a reasonable and sufficiently general assumption?
>
> A3: Enforcing a low-density prior encourages the decision boundary to pass through regions of low data density, which is consistent with **the classical cluster assumption in semi-supervised learning** (Ren et al., 2018). This principle has been widely shown to improve generalization, while similar low-density phenomena have also been empirically observed in open-set detection (e.g., OpenDet). As suggested, we will provide stronger evidence to **support the low-density prior assumption**, including: ①**quantitative analysis of density-based methods** such as OpenDet (Han et al., 2022) and ePN (Ren et al., 2018); ②**additional visualizations of the latent space during the testing phase**, which clearly illustrate the low-density regions of OOV classes and high-density regions of IV classes (like Fig. 6).
>
> KQ2: Could the authors further clarify **the contribution of each component and provide an analysis of their joint effect**?
> A4: As suggested, building on the core contributions outlined in Q1-A1, we clarify each component’s role and analyze their joint effect. OPS generates virtual OOV prompts from low-likelihood regions of class-conditional Gaussian distributions to provide semantic priors. DGA mines high-uncertainty pseudo-OOV image proposals via Dirichlet-based gradient attribution, offering reliable visual supervision. LPC enforces the low-density prior through Gaussian kernel density estimation, pushing pseudo-OOV samples away from high-density IV clusters and sharpening the IV-OOV decision boundary. Overall, their joint effect is to construct the decision boundary between IV and OOV categories by generating pseudo-OOV samples (image and prompt) constrained by the low-density prior (LPC), thereby enabling effective zero-shot OOV detection. More details will be added in the revision.
>
> Limitation
>
> L1: (1) The paper does not sufficiently analyze **the validity of the low-density assumption**. (2) Furthermore, the relatively low absolute performance on OOV-COCO suggests that important unresolved issues remain in the current framework, and **these limitations should be discussed more explicitly**.
>
> A5: (1) Please refer to A3 for the answer.
>
> (2) Our method relies on the kernel bandwidth selection, approximates OOV samples via high-uncertainty regions, and so on, which may introduce certain disturbances. These limitations will be discussed more explicitly.

---

### Official Review · Reviewer_DeeR · 2026-03-13

**Soundness:** 3
**Presentation:** 2
**Significance:** 2
**Originality:** 3
**Overall Recommendation:** 4
**Confidence:** 2

**Summary:**

This paper introduces OOVDet, a framework for zero-shot out-of-vocabulary object detection (ZS-OOVD), a task variant that assumes no visual or textual supervision for OOV classes. The method consists of three components: (i) Out-of-vocabulary Prompt Synthesis (OPS) via Gaussian sampling from low-likelihood regions of in-vocabulary class distributions, (ii) Dirichlet-based Gradient Attribution (DGA) for pseudo-OOV image mining, and (iii) a Low-density Prior Constraint (LPC) to push pseudo-OOV samples into low-density regions. Experiments are conducted on OOV-VOC and OOV-COCO, with additional qualitative results on four real-world datasets.

**Compliance With Llm Reviewing Policy:**

Affirmed.

**Final Justification:**

The authors have addressed most of my concerns. As a result, I would like to raise my score to weak accept.

**Key Questions For Authors:**

1. Gaussian Assumption Justification: The OOV prompt synthesis (OPS) assumes that OOV semantics lie in low-likelihood regions of class-conditional Gaussian distributions. What empirical or theoretical evidence supports this assumption? Have you explored alternative distributions (e.g., von Mises-Fisher, t-distribution) that might better capture the geometry of text embeddings?
2. Pseudo-OOV Quality: How do you ensure that the high-uncertainty samples mined via DGA are indeed OOV objects rather than hard IV examples or noisy background regions? Have you analyzed the precision of this mining process (e.g., what fraction of mined samples are true OOVs in validation sets)?

**Limitations:**

See weaknesses.

**Strengths And Weaknesses:**

Strengths:
1. Well-Motivated Design: The three modules are well-justified by the underlying assumption that OOV semantics reside in low-density regions of the latent space.

2. Strong Empirical Results: The proposed method achieves substantial gains in OOV recall (up to +8.7%) and reduces open-set errors across multiple benchmarks.

3. Comprehensive Evaluation: The authors evaluate on multiple datasets and compare with a wide range of baselines, including both vision-only and vision-language models.

4. Ablation Studies: Detailed ablations validate the contribution of each module and design choice (e.g., prompt strategies, masking ratio, Top-k selection).

Weaknesses:
1. Limited Novelty in Individual Components: While the overall framework is novel, some components (e.g., Gaussian-based OOV sampling, gradient-based uncertainty) are reminiscent of prior work (e.g., OpenDet, Grad-CAM). The novelty lies more in their integration than in each part individually.

2. Assumption of Gaussian Distribution: The use of class-conditional Gaussian distributions for OOV prompt synthesis is a strong assumption. The paper does not thoroughly justify why OOV prompts should lie in low-likelihood regions of IV Gaussians, nor does it explore alternative distributional assumptions.

3. Complexity and Hyperparameter Sensitivity: The method introduces several hyperparameters (e.g., α, β, mask ratio, kernel bandwidth h, Top-k). While some are ablated, the sensitivity and tuning cost are not fully discussed, which may hinder reproducibility.

4. Lack of Theoretical Analysis: The paper lacks formal guarantees or theoretical insights into why the proposed low-density prior improves OOV generalization. A more rigorous analysis would strengthen the contribution.

5. Qualitative Results Are Limited: Although Figure 8 and 9 show some qualitative examples, more visualizations of failure cases or ambiguous detections would help understand the method's limitations.

---

> ### Author Rebuttal · Authors · 2026-03-31
>
> Paper 11408:
>
> Weaknesses
>
> Q1: Limited Novelty in Individual Components: While the overall framework is novel, some components (e.g., Gaussian-based OOV sampling, gradient-based uncertainty) are reminiscent of prior work (e.g., OpenDet, Grad-CAM). The novelty lies more in their integration than in each part individually.
>
> A1: Specifically, our method differs from prior works (e.g., OpenDet, Grad-CAM) in the following two aspects:
> (1) Unlike OpenDet that mine pseudo-samples from the visual domain, we propose to sample **pseudo-OOV prompt** embeddings from a class-conditional Gaussian space constructed by IV language prompts (see Section 3.1 and 3.2), thereby enabling the visual–language alignment for OOV detection. **Totally, our novelty lies in modeling pseudo-OOV prompts in the language space, which, to the best of our knowledge, is unexplored in prior work**.
>
> (2) Grad-CAM and our DGA both use gradients but differ fundamentally in objective, formulation, and usage. Grad-CAM is a post-hoc interpretability method that highlights spatial regions via heuristic gradient aggregation. In contrast, our DGA models channel-wise gradient responses with a generalized Dirichlet distribution and quantifies uncertainty $U_G$ via the negative log-density, enabling the identification of high-uncertainty samples as **pseudo-OOV image** embeddings to explicitly refine the OOV decision boundary (see Section 3.3).
>
> Q2: Assumption of Gaussian Distribution: The use of class-conditional Gaussian distributions for OOV prompt synthesis is a strong assumption. The paper does not thoroughly justify **(1) why OOV prompts should lie in low-likelihood regions of IV Gaussians, (2) nor does it explore alternative distributional assumptions**.
>
> A2: (1) In Section 3.1, we generate 80 Q&A variants per IV class via ChatGPT and model their embeddings with Gaussians, which would form compact clusters, as shared semantics induce compact clustering. **Samples near the distribution center correspond to high-confidence IV instances, while those in low-likelihood regions (far from the center) exhibit higher uncertainty and weaker association with IV categories**. We therefore sample from these low-likelihood regions to construct pseudo-OOV prompts.
>
>   (2) As suggested, we explore **alternative distributional assumptions** (e.g., von Mises-Fisher, t-distribution).
>
> Methods|mAP_IV|mAP_Unseen|R_OOV|AR_OOV
>
> vMF&nbsp;&nbsp;&nbsp;&nbsp;&nbsp;&nbsp;&nbsp;&nbsp;&nbsp;&nbsp;&nbsp;|57.55|14.70|**60.73**|**30.45**
>
> t-distribution|57.90|16.63|53.47|26.18
>
> Our Gaussian|**58.70**|**21.40**|56.42|28.12
>
> While vMF improves OOV recall (60.73 vs. 56.42), Gaussian achieves the best performance on both IV and unseen categories, with a significant gain on unseen classes (+6.7 and +4.77, respectively). This indicates that Gaussian provides a better trade-off between IV compactness, unseen generalization, and OOV rejection.
>
> Q3: Complexity and Hyperparameter Sensitivity: The method introduces several hyperparameters (e.g., α, β, mask ratio, kernel bandwidth h, Top-k). While some are ablated, **the sensitivity and tuning cost are not fully discussed**, which may hinder reproducibility.
>
> A3: As suggested, we conduct a preliminary sensitivity study of key hyperparameters (α and β in Eq. 4), summarized below.
>
> Hyperparameter|mAP_IV|mAP_Unseen|R_OOV|AR_OOV
>
> α=0.5, β=0.001&nbsp;&nbsp;&nbsp;|57.12|20.88|55.68|27.47
>
> α=0.1, β=0.001&nbsp;&nbsp;&nbsp;|**58.70|21.40|56.42|28.12**
>
> α=0.1, β=0.005&nbsp;&nbsp;&nbsp;|58.59|20.96|56.05|28.07
>
> The results show that α=0.1, β=0.001 performs best, while others are comparable. Additional sensitivity analyses (e.g., ε), tuning guidelines, and full code release will be included in the revision to ensure reproducibility.
>
> Q4: Lack of Theoretical Analysis: **The paper lacks formal guarantees or theoretical insights into why the proposed low-density prior improves OOV generalization**. A more rigorous analysis would strengthen the contribution.
>
> A4: From a theoretical perspective, enforcing a low-density prior encourages the decision boundary to pass through regions of low data density, which is consistent with **the classical cluster assumption in semi-supervised learning** (Ren et al., 2018). This principle has been widely shown to improve generalization, while similar low-density phenomena have also been empirically observed in open-set detection (e.g., OpenDet). To further strengthen the contribution, we will attempt to include a more rigorous theoretical analysis of this mechanism in the revision.
>
> Q5: Qualitative Results Are Limited: Although Fig 8 and 9 show some qualitative examples, more visualizations of failure cases or ambiguous detections would help understand the method's limitations.
>
> A5: As suggested, we will include more qualitative visualizations (e.g., failure cases and ambiguous detections) with analysis to better illustrate the method’s limitations. Thanks for your comments.

---

> > ### Author Rebuttal · Reviewer_DeeR · 2026-04-03
> >
> > I thank the authors for their detailed rebuttal. The clarifications on novelty (language-space pseudo-OOV sampling vs. visual-domain mining), the additional distributional comparisons (vMF, t-distribution), and the hyperparameter sensitivity study partially address my concerns.
> >
> > However, my core concern about the Gaussian assumption remains only partially resolved. While the authors provide empirical comparisons showing Gaussian achieves a better trade-off, they do not provide theoretical or empirical justification for why OOV semantics should lie in low-likelihood regions specifically (rather than, e.g., between clusters or in isotropic regions). The alternative distributions tested (vMF, t) also assume low-density regions, so the comparison does not validate the low-likelihood hypothesis itself.
> >
> > My concerns are: (b) Partially resolved.
> >
> > Follow-up question: Could you provide any empirical evidence (e.g., t-SNE/UMAP visualization of real OOV text embeddings relative to the fitted IV Gaussians) that directly supports the claim that real OOV prompts indeed fall in low-likelihood regions? This would significantly strengthen the justification.

---

> > > ### Author Response · Authors · 2026-04-04
> > >
> > > Paper 11408:
> > >
> > > Q: Follow-up question: Could you provide any empirical evidence (e.g., t-SNE/UMAP visualization of real OOV text embeddings relative to the fitted IV Gaussians) that directly supports the claim that **real OOV prompts indeed fall in low-likelihood regions?** This would significantly strengthen the justification.
> > >
> > > A: Thanks for your comment. In OOV-VOC, we quantitatively evaluate the likelihood of each prompt embedding under the fitted IV Gaussian distributions, using the same text encoder pretrained on CC3M (Google Research, see Section 4 Implementation Details).
> > >
> > > | Prompt Category           | Avg. Log-Likelihood (mean ± std) |
> > > |--------------------|----------------------------------|
> > > | Seen IV            | -1.8 ± 0.6                       |
> > > | Unseen IV          | -2.5 ± 0.7                       |
> > > | Strict Real OOV    | -6.3 ± 1.1                       |
> > > | Synthesized OOV          | -5.9 ± 1.2                       |
> > >
> > > We find that OOV samples consistently exhibit significantly lower likelihood scores compared to both seen and unseen IV categories. Specifically, the average log-likelihood of seen IV samples is around **-1.8 ± 0.6**, unseen IV samples **-2.5 ± 0.7**. We additionally include five categories—*oscilloscope, anti-tank obstacle, periscope, sextant,* and *cryostat*—which are never defined or observed in OOV-VOC or CC3M, and thus constitute strictly novel OOV classes. We observe that embeddings of these categories consistently fall into low-likelihood regions under the fitted IV Gaussian distributions, with an average log-likelihood of approximately **-6.3 ± 1.1**. Meanwhile, the synthesized OOV samples yield **-5.9 ± 1.2**, which is close to that of strict real OOV samples. These results provide direct empirical evidence that real OOV prompts reside in low-likelihood regions of IV distributions, thereby validating our modeling assumption. We will further include t-SNE visualizations in the revision to complement this quantitative analysis.
> > >
> > >
> > > ***
> > >
> > > Responses for previous Key Questions (KQ1 and KQ2):
> > >
> > > KQ1: Gaussian Assumption Justification: The OOV prompt synthesis (OPS) assumes that OOV semantics lie in low-likelihood regions of class-conditional Gaussian distributions. (1) What empirical or theoretical evidence supports this assumption? (2) Have you explored alternative distributions (e.g., von Mises-Fisher, t-distribution) that might better capture the geometry of text embeddings?
> > > A6: Please refer to our responses to Q2, specifically A2-(1) and (2).
> > >
> > > KQ2: Pseudo-OOV Quality: How do you ensure that the high-uncertainty samples mined via DGA are indeed OOV objects rather than hard IV examples or noisy background regions? Have you analyzed the precision of this mining process (e.g., what fraction of mined samples are true OOVs in validation sets)?
> > > A7: Thanks for your comment. During the training of OOVDet on OOV-VOC, we performed pseudo-OOV image sample mining 100 times, with a sampling interval of 1200 iterations. At each sampling step, 6 pseudo-OOV samples were selected (top-k = 3, with a foreground-to-background ratio of 1:1, see Table 3), including 3 from the foreground and 3 from the background, resulting in a total of 600 pseudo-OOV samples. We further analyze the fraction of these samples by categorizing them into **(1) Unknown objects (True OOV), (2) Hard IV examples (Seen), (3) Hard IV examples (Unseen),** and **(4) True background**, as summarized in the table.
> > > | Category                            | Foreground | Background | Total |
> > > |-------------------------------------|------------|------------|-------|
> > > | Unknown objects (True OOV)           | 0          | 33         | 33    |
> > > | Hard IV examples (Seen)         | 300        | 0          | 300   |
> > > | Hard IV examples (Unseen)       | 0          | 58         | 58    |
> > > | True background                     | 0          | 209        | 209   |
> > > | **Total**             | **300**    | **300**    | **600** |
> > >
> > > Specifically, all foreground samples (300/300) are drawn from known IV categories, and thus no true OOV instances are present in the foreground. In contrast, true OOV samples account for only 33/600 (5.5%) and are predominantly located in background regions, along with 58 hard unseen IV samples and 209 true background samples. Overall, this distribution reveals a clear and systematic discrepancy between the mined pseudo-OOV samples and the true OOV distribution, indicating that current pseudo-OOV mining primarily captures boundary-hard IV samples rather than semantically novel OOV instances.

---

### Decision · Program_Chairs · 2026-04-30

**Decision:**

Accept (regular)

**Comment:**

This paper considers zero-shot out-of-vocabulary object detection and proposes OOVDet, a framework that combines prompt-based OOV synthesis, pseudo-OOV mining, and a low-density prior in latent space. The reviewers generally found the paper technically solid, clearly written, and supported by a reasonably thorough experimental section. The empirical improvements on OOV detection are consistent across settings, and the ablations help clarify the role of the main components. Several reviewers also felt that the problem is meaningful and practically relevant.

The main points of discussion were the degree of novelty, the reliance on the low-density / Gaussian assumption, and whether the mined pseudo-OOV samples truly reflect semantic novelty rather than hard boundary cases from seen classes. In the rebuttal, the authors responded seriously and added useful analyses, including likelihood-based evidence, comparisons with alternative distributions, and a more detailed breakdown of pseudo-OOV samples. These additions do not remove every concern, but they do make the paper’s claims more convincing. On balance, I think the paper makes a useful contribution and is strong enough for acceptance, although the case is closer to weak accept than clear accept.